# Postpartum Relapse in Patients with Bipolar Disorder

**DOI:** 10.3390/jcm11143979

**Published:** 2022-07-08

**Authors:** Javier Conejo-Galindo, Alejandro Sanz-Giancola, Miguel Ángel Álvarez-Mon, Miguel Á. Ortega, Luis Gutiérrez-Rojas, Guillermo Lahera

**Affiliations:** 1Psychiatry Department, Institute of Psychiatry and Mental Health, Hospital General Universitario Gregorio Marañón, 28007 Madrid, Spain; doctorjcgalindo@gmail.com; 2Alcalamente, Mental Health Service, 28806 Alcalá de Henares, Spain; 3Psychiatry Department, Principe de Asturias University Hospital, 28805 Alcalá de Henares, Spain; sirgiancola@gmail.com (A.S.-G.); guillemo.lahera@gmail.com (G.L.); 4Department of Medicine and Medical Specialities, Faculty of Medicine and Health Sciences, University of Alcalá, 28801 Alcalá de Henares, Spain; maalvarezdemon@icloud.com (M.Á.Á.-M.); miguel.angel.ortega92@gmail.com (M.Á.O.); 5Department of Psychiatry and Mental Health, Hospital Universitario Infanta Leonor, 28031 Madrid, Spain; 6Ramón y Cajal Institute of Sanitary Research (IRYCIS), 28034 Madrid, Spain; 7Department of Psychiatry, Faculty of Medicine, University of Granada, 18016 Granada, Spain; 8Biomedical Research Networking Centre in Mental Health (CIBERSAM), 28029 Madrid, Spain

**Keywords:** bipolar disorder, postpartum, treatment, management, relapse, prevention

## Abstract

Pregnancy and postpartum are vital times of greater vulnerability to suffer a decompensation of bipolar disorder (BD). Methods: A systematic literature search was performed on public electronic medical databases, following PRISMA guidelines. Studies were included if they reported postpartum relapse in patients diagnosed with BD according to Diagnosis Statistical Manual (DSM) or International Classification Disease (ICD) criteria. Results: Sixteen articles describing 6064 deliveries of 3977 women were included in the quantitative analyses. The overall risk of postpartum relapse was 36.77%. The methodology of the studies, the diagnostic criteria, the discrimination between BD type I and II, and the origin of the sample were very heterogeneous. Conclusions: the rate of postpartum bipolar relapse is very high, as it is considered to be a critical period. It is especially important to detect decompensation in this period and to evaluate mood-stabilizing treatment, given the high risk of relapse concentrated in a short period.

## 1. Introduction

Bipolar Disorder (BD) is a severe affective mood disorder characterized by a wide range of lifelong mood changes, varying between depressive, hypomanic, or manic, with or without mixed features [1,2]. Because of the average age of onset of BD, many women will face their reproductive desires with the threat of the risks inherent to this pathology. Pregnancy is a critical period, both physiologically and emotionally, with an increased likelihood of relapse. Although much uncertainty still remains about the risk of mood episodes during pregnancy [3], women with a history of BD are at high risk for postpartum relapse [4].

Postpartum is a period of high risk for the appearance and recurrence of psychiatric disorders, particularly depression, mania, and psychoses [5]. In a classical population-based cohort, patients with BD had a postpartum risk of psychiatric admission due to relapse of 16%, much higher than that of other mental disorders (3% for patients with schizophrenia and 2% for patients with major depressive disorder) [6]. Munk-Olsen et al. [7] analyzed the risk of perinatal psychiatric readmission in a population of more than 28,000 women with preexisting mental illness, finding a higher relative risk of relapse during the early postpartum period for BD patients (relative risk = 37.2, 95% CI = 13.6, 102.0) compared to schizophrenia patients (relative risk = 4.6, 95% CI = 2.5, 8.5) or other psychiatric disorders (relative risk = 3.0, 95% CI = 1.9, 4.7). The period of highest risk of hospitalization for relapse in women with BD was between days 10 and 19 postpartum.

Episodes of hypomania, mania, or depression may occur in the context of BD or develop for the first time after childbirth. Although not as prevalent as the baby blues, elation and associated hypomanic symptoms (generally referred to as the “highs”) are also common after childbirth [8]. It is important to detect these “highs”, even if they do not have a functional impact, because these postpartum hypomanic symptoms may be associated with subsequent depression [8].

Childbirth can also be the trigger for bipolar depression. Of all women diagnosed with postpartum depression, 54% met the diagnostic criteria for BD, but only 10% of those had a previous diagnosis [9], which suggests the need for screening. These depressions usually have diurnal variations in mood, hypersomnia, hyperphagia, distractibility, and irritability. Therefore, BD depression should be considered postpartum in women with atypical depressive symptoms, with a family history of BD, or who do not respond to antidepressant treatment [10]. When these patients are being treated with antidepressants, they may present mixed features, in which the probability of suicide is higher, so close monitoring is required [11].

Significant hormonal changes and physiological changes after childbirth can act as a trigger for relapse in these patients [12]. In many cases, the psychopharmacological treatment of the patient is suspended during pregnancy, due to the teratogenic risk, which depends on the characteristics of the drug and the time of exposure [13].

Underestimation of relapse risk could lead to ineffective prevention strategies and delayed referral for specialized perinatal care, could impair quality of life, and could cause a higher rate of acute hospitalization and suicide [14,15]. The relevance of this issue transcends into the planning of subsequent pregnancies. Women with first-onset postpartum mania or psychosis are less likely to have more children, and, furthermore, women with BD are known to have lower fertility rates compared to the general population [16]. The likelihood of obstetric complications (such as gestational hypertension and antepartum hemorrhage, labor induction, cesarean section) and complications to the fetus (low birth weight, increased risk of malformations, increased neonatal morbidity) is also higher in this population [17].

All these data make us consider the critical moment that the postpartum period represents in women with a history of BD, which motivates the need to know the approximate risk of relapse and to adopt subsequent management and prevention measures at different levels. The present review offers a summary of the available scientific literature on the relapse risk in women with BD postpartum and the principal factors associated with this phenomenon.

## 2. Materials and Methods

This review followed the Preferred Items for Reporting of Systematic Reviews and Meta-Analyses (PRISMA) guidelines [18].

### 2.1. Subsection

The inclusion criteria for the studies were as follows: (1) quantitative or qualitative research aimed at examining the influence of postpartum on decompensations in patients with previous BD, (2) English-language publications, and (3) participation of patients of all ages. Exclusion criteria were: (1) review articles and meta-analyses, (2) articles that included patients with other diagnoses in addition to BD and did not separate the results according to such diagnoses, and (3) papers with patients who did not meet to Diagnosis Statistical Manual-IV (DSM-IV) [19] or International Classification Disease-10 (ICD-10) [20] criteria for BD diagnosis.

### 2.2. Search Strategy

The search was realized until 7 January 2022 in the following databases: PubMed, PsycINFO, and Web of Science. The search strategy used in each of these databases was as follows: “bipolar disorder” OR “manic depressive illness” AND (postpartum OR delivery, obstetric OR parturition OR abortion, spontaneous OR abortion, induced). The filters applied in the three databases met the inclusion criteria.

### 2.3. Study Selection Process

This process was carried out in four phases. First (article identification phase), the results of the searches in the three databases were unified, and duplicate articles were eliminated. Second (screening phase), the titles and abstracts of articles that potentially met the inclusion criteria were read. If there were doubts, we proceeded to full-text review of the doubtful article. Third (eligibility phase), the full-text articles were prescreened in the previous phase, and the doubtful articles were independently reviewed and read. Finally (inclusion phase), the articles included in the present systematic review were finally selected. The four phases were realized for two coauthors (J.C. and A.S.-G.)

### 2.4. Data Extraction Process for Each Study

The following information was extracted from the selected articles: (1) title of the study, author/s, and year of publication, (2) size of the patient sample, (3) characteristics of the participants (sociodemographic data, diagnosis, whether they were outpatients or inpatients and phase of the disease at the time of assessment of bipolar relapse), (4) characteristics of the study (methodology), (5) type of treatment received by the patients, (6) measurement of bipolar relapse (measuring instrument, type of assessor, and time of measurement of bipolar decompensation), and (7) measurement of results (measuring instrument and type of assessor).

## 3. Results

Initially, 893 original studies resulting from the application of the search strategy described above were selected. Figure 1 shows the selection process of these studies. After following the recommendations of the PRISMA guidelines, the 16 articles that met the inclusion criteria and did not meet any of the exclusion criteria were selected. The main characteristics of the selected articles are summarized in Table 1. The 16 chosen studies included a total of 3977 women who had a previous diagnosis of BD at the start of each study, of whom a total of 6064 deliveries were counted. Although all participants had a previous diagnosis, the studies included in the systematic review reevaluated them using internationally validated criteria and/or instruments (DSM, ICD, SCAN, SCID, RDC, HRS, YMRS, BRMS, SADS). Of the 16 studies, 10 were retrospective [21,22,23,24,25,26,27,28,29,30], 5 were prospective ([15,31,32,33,34], with 1 observational (31) and 1 naturalistic (33)), and 1 was a single-blind clinical trial [35]. The main variable extracted from the studies was the risk of postpartum relapse. Unifying all the results, the overall risk of postpartum relapse was 36.77%.

### 3.1. Sociodemographic Characteristics

The selected studies included women aged between 18 and 44 years (we did not detect articles with patients outside this age range). Based on the data available in nine studies [15,21,25,26,28,30,33,34,35], the weighted mean age of the patients was 26–30 years at the time of delivery. The rest of the analyzed variables, as well as the other sociodemographic characteristics of the participants, turned out to be very heterogeneous: both single women and women in a couple, with medium or higher education, working or unemployed, of different ethnicities and countries, with varying socioeconomic levels, smokers and nonsmokers, with planned and unplanned pregnancies. Some authors even dare to conclude that psychosocial factors do not play a significant role in perinatal episodes.

### 3.2. Clinical Characteristics

The most important clinical variable analyzed in this systematic review is the type of BD. Considering the data from the 13 studies that differentiated between type I and II BD [15,21,23,24,25,26,29,30,31,32,33,34,35], the weighted mean percentage of patients with BD type I was 69.83%. One of the studies included in the review segregated the results into two different types of category: BD type I/schizoaffective BD and BD type II/BD not otherwise specified, and related disorders [34]. The studies generally analyze very heterogeneous clinical factors, although there are some common variables among the studies that are of interest, such as the age of onset of the illness [21,22,23,24,25,26,29,30,31,32], family history of affective disorder [25,29,30,32,33], duration of the illness [22,26,30,32,34], presence of episodes in the past [29,30,32,34], previous perinatal episodes (depending on the article, we will speak specifically of late pregnancy, postpartum, and/or puerperium) [15,25,26,31,33,34,35], rapid cycling [29,32,35], the use of prophylaxis with mood-stabilizers and/or antipsychotics in the perinatal period [15,23,25,28,29,31,33,34,35], alcohol abuse [32,35], previous suicide attempts [24,25,32,35], or the number of hospitalizations [25,31,33,35].

### 3.3. Subtype Episode

Studies generally differentiate between manic and depressive episodes, although some differentiate specific categories of hypomanic [24,25,26,30,32] and/or mixed features (24–26,30,35). When the type of episode was specified, four studies coincided in pointing out the existence of a greater probability of suffering episodes of postpartum psychosis/mania in women with a previous diagnosis of BD I [21,24,29,34]. Considering all types of BD, depression was the most prevalent form of postpartum morbidity [24,26,32].

### 3.4. Pregnancy and Childbirth

The most studied clinical factor related to pregnancy was the number of pregnancies per woman. Most studies include, in their sample, women with a variable number of pregnancies, with only one study limiting the sample to primiparous women [27]. A postpartum relapse in the first pregnancy usually leads to a decreased likelihood of having more children [36] and to an increased risk of new episodes in subsequent pregnancies [24,29].

### 3.5. Factors Associated with Postpartum Relapse

#### 3.5.1. Sociodemographic Factors

On the association between sociodemographic factors and postpartum relapse, most studies agree that there is no greater prevalence of specific characteristics in relapse episodes. There are only very few works venturing to make points in this regard, such as those who claim that the risk of relapse is more likely in women with a younger age at delivery [25] or with a smaller number of children and a higher academic level [26].

#### 3.5.2. Clinical Factors

Most studies point out that risk of relapse is more prevalent in perinatal episodes having a lower-than-average age at the onset of the disease [22,26,29,30], having a longer duration of the episode [30,32], having suffered previous episodes of the disease in the past, having a family history of BD [25,29,30], or having had psychiatric hospital admissions in the period prior to pregnancy, especially if there have been several and/or recent episodes [27]. The risk of relapse was found to be higher during the postpartum period than during pregnancy in women diagnosed with BD [26,29,31], presenting similar risk patterns for a perinatal episode in the two diagnostic groups (type I and II) [26,34]. One study points out that this risk is even higher after live birth compared to miscarriage and induced abortion [29], although the chosen method of delivery could not predict the risk of relapse [21,22]. A situation of particular risk occurs when, during the period of pregnancy, no treatment is received [24]. If we add to this that the rate of postpartum relapse in women with BD was higher when they experienced previous episodes during pregnancy [15,26,34] or during first gestation [23,24], perinatal prophylaxis in BD becomes essential to reduce the risk of relapse [15,23,28]. Other predictors of postpartum relapse, such as obstetric complications during pregnancy [22], unplanned pregnancies, or poor disease remission during gestation, have also been reported [25].

## 4. Discussion

In this analysis, we included 16 studies that analyzed a total of 6064 deliveries of women with previous BD diagnosis, finding an overall relapse risk of 36.77%. Studies with a larger sample size detected intermediate risks, as analyzed studies with more than 1000 deliveries [21,26] obtained higher relapses rates of 36–45%. This is a very high risk for a very specific short period of time, which underlines the importance of this critical period. It has been suggested that this may be due to the increase in placental hormones during pregnancy and their abrupt cessation after birth [37] and other biological factors [12], as well as being an important psychosocial stressor [26].

Despite the interest in these findings in the scientific literature, few studies have analyzed the risk of relapse accurately and with quality criteria. On the other hand, many studies focus on general recommendations for childbirth and postpartum, as well as on treatment during pregnancy. In recent years, there has been a high scientific production in relation to the postpartum period in patients diagnosed with BD; however, they have not focused so much on the intrinsic risk of relapse as on recommendations on prophylactic treatment during pregnancy [38,39,40] or on clinical characteristics of these new episodes [8,41]. These recommendations are occasionally conditioned by factors, such as the severity of illness or pregnancy plans [42].

We found great heterogeneity in these studies, with the following being the main factors to highlight:−Terminology: In the oldest studies, there is terminological confusion when speaking generically of postpartum psychosis [43], even using the term to encompass relapses in BD. Therefore, older studies could not be included [6,44,45,46,47].−Retrospective studies: These are the most abundant, with biases that can be inferred from this methodology, as they certainly do not detect all relapses. Of the 16 studies selected, only 5 were not retrospective, which additionally presented a reduced sample of patients (201 patients in total).−Duration of the postpartum period: This is highly variable among the different studies, ranging from 1 to 9 months, which can have a determining influence on the results of relapses. However, studies show that most relapses are early [48]. A broader temporal definition may be beneficial in clinical practice to cover care for mothers who experience a relapse in the following months after delivery. However, a broad definition may be problematic for studies examining the mechanism of postpartum relapse.−Type of BD: Studies with type I and II BD have been included indistinctly, but the variability in the proportion of the sample is wide, from studies that only include type I [23] or only type II [31] to several studies in which there is no differentiation whatsoever.−Relapse diagnosis: This is highly conditioned by the type of retrospective study, as most of them used a retrospective assessment according to DSM or ICD criteria. Hypomanic episodes are particularly difficult to detect retrospectively from the available documentation. In that sense, studies with a more complete detection are prospective and use specific scales for BD status assessment. However, they have the bias of including the use of certain treatments as the main variable (olanzapine and valproate, respectively) [33,35].−Medication use: The choice of medication strategies is heterogeneous. There is no modification in retrospective studies (and in most of them, there is no analysis of these treatments), although there are some studies in which the usual treatment is modified. Analyzing the results of our review and of recent works that have studied the possible teratogenic effects of mood stabilizers, we believe that the most convenient thing to do is to continue using lithium treatment at the lowest possible dose (especially during the first trimester of pregnancy) [49], while valproic acid is completely contraindicated [50].−Sample selection: Most of the studies use samples from specialized BD programs instead of community samples.−Sample size: Most studies collected small samples. Only eight studies present a sample size greater than 100, and only two greater than 1000 deliveries [21,26]. These two studies are important and condition the final results. Di Florio et al. [21] conducted a retrospective study, but they not only relied on patient history but also complemented it with patient interviews. In this study, no considerations were made about the treatment performed. The sample obtained from cohorts was included in genetic studies of affective disorders, which could lead to a selection bias. In their results, they highlight that primiparous women with BD type I are those with the highest risk of postpartum relapse. Subsequently, Di Florio et al. were able to demonstrate that previous prenatal history of affective psychosis or depression is the most important predictor of perinatal recurrence in women with bipolar disorder [51]. Viguera et al. [26] showed a risk of relapse in mothers with BD of 3.5-times higher during postpartum than during pregnancy. The participants were selected from a Perinatal Psychiatry program, with a lower incidence of relapse than the previous study (36% vs. 45%), despite considering the postpartum for 6 months, instead of the 6 weeks considered by Di Florio. The existence of greater severity in the sample collected by this author could be considered.−Pregnancy planning: The total rate of unplanned pregnancies in BD is around 50% [52]. The risk of relapse, as well as the other risks of treatment in pregnancy, should be discussed with all women of childbearing age with mood disorders, even those who are not planning a pregnancy. Because women may not be in contact with psychiatric services, it is important that all professionals providing medical care to pregnant women, including midwives, family physicians, and obstetricians, are aware of this increased risk.

Based on our findings, there should be special attention and close follow-up during the postpartum period, as well as special emphasis on treatment—In particular, restarting treatment in those women who had discontinued treatment during pregnancy. Pregnant women have multiple contacts with health services, which provides an opportunity regarding BD for supervision during pregnancy and closer contact in the postpartum period. Although the definition of the postpartum period is not homogeneous in the various studies, the highest risk of relapse occurs in the days and weeks closest to delivery.

More prospective studies are needed with large samples, without inferences from trials that include medication changes, and samples not exclusively selected from specific programs that infer biases due to greater severity of the disorder.

### Strengths and Limitations

To the best of our knowledge, this systematic review includes the most up-to-date data concerning the risk of relapse in BD. Systematically, the literature was reviewed up to and including 7 January 2022.

As limitations, the heterogeneity of some variables, such as the origin of the samples, the different treatment strategies, and the different definitions used for the postpartum period, have prevented a quantitative analysis of the studies in the form of a meta-analysis, but, from the articles, we can infer descriptive statistics of important interest due to their clinical repercussions.

Apart from the observation of relapse in the postpartum period, the review includes the risk factors related to the event studied. One of the strengths of the study is that risk factors were collected and stratified according to the multiple stages of the woman’s life, starting with her sociodemographic factors (controversial variables, little considered, but, nevertheless, sometimes presenting statistical significance), followed by the debut of the disease, continuing during the course of pregnancy, and ending in the perinatal period. This is a novelty because other recent synonymous systematic reviews were concerned with the identification of factors associated with relapse, but they focus on specific phases, such as the review by Rusner et al. [17], which focused specifically on the period of pregnancy and birth, and specifically, on adverse medical events in the mother and child. Recent work has been published that focuses on identifying risk factors at different stages of a woman’s life, although these are non-systematic narrative reviews [53]. Other systematic reviews focus on making important nosological distinctions, such as the difference in the risk of relapse between women with BD and women with a history of postpartum psychosis [4], or the importance of the phenomenon of hypomania, or high, in the postpartum period in women [40].

## 5. Conclusions

To facilitate access to the factors described, we propose, in Figure 2, a conceptualization model for understanding postpartum relapse, from both a cross-sectional and longitudinal perspective. Some of these variables share scientific evidence from several studies; however, there are others, particularly sociodemographic variables, that have statistical significance in studies with a large sample size, but they should be further studied for their potential clinical implications.

## Figures and Tables

**Figure 1 jcm-11-03979-f001:**
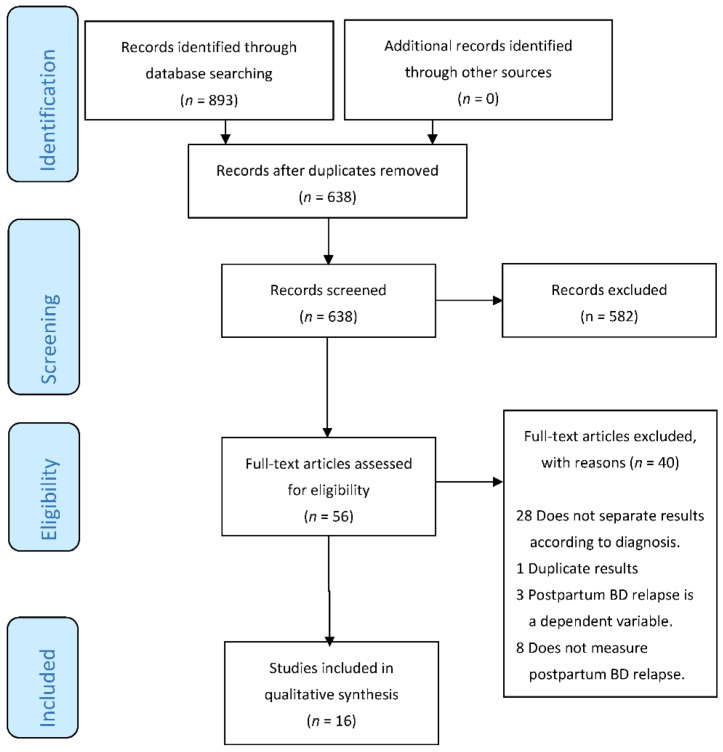
Preferred Reporting Items for Systematic Reviews and Meta-Analyses (PRISMA) flow diagram.

**Figure 2 jcm-11-03979-f002:**
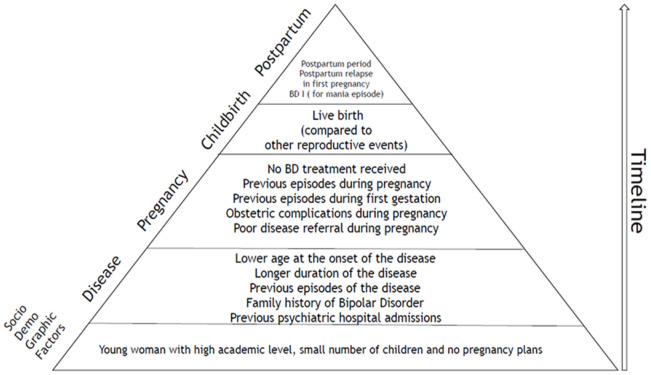
Risk factors for postpartum relapse in patients with bipolar disorder extracted from the selected articles.

**Table 1 jcm-11-03979-t001:** Results of studies estimating the prevalence of postpartum relapse in patients with bipolar disorder.

	Patients			Origin of the Sample	Type of Study	Methods		Results
Study	N	Mean Age of Pregnant (SD)	BD I %			Instruments	Definition of the Relapse	% Relapse (TOTAL)
(Di Florio et al., 2014) [21]	1.212	25.00 (5.25)	77.1%	Participants with affective disorders	Retrospective	SCAN DSM-IV	6 weeks	45% (1.052/2.329)
(Akdeniz et al., 2003) [22]	72	NA	NS	Analysis of consecutive cases	Retrospective	SCID	26 weeks	16.3% (26/160)
(Grof et al., 2000) [23]	28	NA	100%	Patients treated with lithium	Retrospective	RDC	36 weeks	25% (7/28)
(Maina et al., 2014) [24]	276	NA	46.7%	Mood and Anxiety Unit	Retrospective	DSM-IV TR	4 weeks	75% (207/270)
(Sharma et al., 2013) [31]	37	NA	0 (100% BD-II)	Study about the use of drugs during the pregnancy	Observational prospective	DMS-IV	4 weeks	70% (26/37)
(Doyle et al., 2012) [25]	43	32 (5.5)	80.8%	Perinatal Mental Health Service	Retrospective	DSM-IV	6 weeks	47% (20/43)
(Bergink et al., 2012) [15]	41	33.6 (3.8)	27%	Preventive postpartum program	Prospective	DSM-IV	4 weeks	22% (9/41)
(Viguera et al., 2011) [26]	621	26.4 (4.9)	45.6%	Perinatal Mental Health Service	Retrospective	DSM-IV	24 weeks	36% (403/1120)
(Colom et al., 2010) [32]	200	NA	70.5%	Bipolar Disorder Program	Prospective	DSM-IV	4 weeks	39% (43/109)
(Harlow et al., 2007) [27]	786	NA	NS	Hospital Discharge and births Sweden Register (1987–2001)	Retrospective	ICD-10	90 days	9% (67/786)
(Sharma et al., 2006) [33]	25	30.3 (6.2)	36%	Preventive study with olanzapine	Naturalistic Prospective	HRS, YMRS, DSM-IV	4 weeks	40% (10/25)
(Wisner et al., 2004) [35]	26	32.4	61.5%	Preventive study with valproate	Simple Blind study	HRS, YMRS, BRMS, DSM-IV	20 weeks	69% (18/26)
(Cohen et al., 1995) [28]	27	33.4 (4.9)	NA	Comparative study with profilactic treatment	Retrospective	DSM-III-R	12 weeks	33% (9/27)
(Perry et al., 2021) [34]	124	BD-I/SAD-BD: 34 (6) BD-II/BD-OS: 32(7)	76.61%	Bipolar Disorder Research Network Pregnancy Study	Prospective	DSM 5 ICD-10 SCAN	12 weeks	BD-I/SA-BD: 40.6% (39/96) BD-II/BD-OS: 44.0% (11/25)
(Gilden et al., 2021) [29]	436	NA	100%	Bipolar cohort from Netherland	Retrospective	SCID	24 weeks	30.1% (277/919)
(Uguz et al., 2021) [30]	23	30.39 (4.42)	69.6%	Comparative study between two antipsychotics (olanzapine and quetiapine)	Retrospective	SCID	32 weeks	26.1% (6/23)
TOTAL	3977	26.30	69.83%					36.77% (2230/6064)

Note: BD I = Bipolar Disorder type I; BD II= Bipolar Disorder type II; BRMS: Bech-Rafaelson Mania Scale; Definition of relapse: bipolar relapse in the first x weeks postpartum; HRS: Hamilton Rating Scale for Depression-31 items; ICD: International Classification Disease; NA: Not Available; RDC: Research Diagnostic Criteria; SAD = Schizoaffective disorder; SADS: schedule for affective disorders and schizophrenia; SCAN: Schedules for Clinical Assessment in Neuropsychiatry; SCID: Structured Clinical Interview for DSM-IV; YMRS: Young Mania Rating Scale.

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
