# Peer review of "Postpartum Relapse in Patients with Bipolar Disorder"

_jcm, 2022, doi:10.3390/jcm11143979_

Round 1

Reviewer 1 Report

In Introduction: “It is important to detect these "highs," even if they do 59 not have a functional impact, because these postpartum hypomanic symptoms may be 60 associated with subsequent depression.” Reference needed.

In  Methods, 2.2. Search strategy, you did not mention which databases were used. You should mention them in detail.

In the PRISMA flow diagram, rearrange excluded results according to frequency. 8 did not measure what? Why didn’t you use the PRISMA 2020 version?

In 3.1. Sociodemographic characteristics “The selected studies included women aged between 18 and 44 years^. This is surprising in view of the fact that you included all ages. This need to be commented. Was it due to the inclusion criteria of the eligible studies?

Table 1. Put the title on the top of the Table.

In 3.5.2. Clinical factors. “in the two diagnosis groups”, “in the two diagnostic groups” is better.

On the whole, quite well written and interesting.

Author Response

In Introduction: “It is important to detect these "highs," even if they do 59 not have a functional impact, because these postpartum hypomanic symptoms may be 60 associated with subsequent depression.” Reference needed.

 Thank you very much, we have attached the next reference (number 8):

Sharma, V., Singh, P., Baczynski, C., Khan, M. (2021). A closer look at the nosological status of the highs (hypomanic symptoms) in the postpartum period. Arch. Womens Ment. Health. 2021; 24, 55-62. https://doi.org/10.1007/s00737-020-01023-1

In Methods, 2.2. Search strategy, you did not mention which databases were used. You should mention them in detail.

We have explained the databases used in the Methods section:

The search was realized until January 7, 2022 in the next databases: PubMed, PsycINFO, and Web of Science.

In the PRISMA flow diagram, rearrange excluded results according to frequency. 8 did not measure what? Why didn’t you use the PRISMA 2020 version?

This is a mistake. Now in the Flow Diagram (Figure 1) we have included the explanation for excluded 8 articles:

8 Does not measure postpartum BD relapse.

The PRISMA reference is a mistake. We have used the 2020 version. Now we have changed the reference to the new one (number 18):

Page, M.J., McKenzie, J.E., Bossuyt, P.M., Boutron, I., Hoffmann, T.C., Mulrow, C.D., Shamseer, L., Tetzlaff, J.M., Akl, E.A., Brennan, S.E., Chou, R., Glanville, J., Grimshaw, J.M., Hróbjartsson, A., Lalu, M.M., Li, T., Loder, E.W., Mayo-Wilson, E., McDonald, S., McGuinness, L.A., Stewart, L.A., Thomas, J., Tricco, A.C., Welch, V.A., Whiting, P, Moher, D. The PRISMA 2020 statement: an updated guideline for reporting systematic reviews. BMJ. 2021; 372, n71.

In 3.1. Sociodemographic characteristics “The selected studies included women aged between 18 and 44 years. This is surprising in view of the fact that you included all ages. This need to be commented. Was it due to the inclusion criteria of the eligible studies?

 In our review of the literature we did not find any studies outside this age range, so we only included studies with women over 18 and under 44 years of age.

We have included an explanation in the Methods section:

The selected studies included women aged between 18 and 44 years (we have not detected articles with patients outside this age range).

 Table 1. Put the title on the top of the Table.

 We have included it on the top of the Table.

 In 3.5.2. Clinical factors. “in the two diagnosis groups”, “in the two diagnostic groups” is better.

We have changed it.

On the whole, quite well written and interesting.

Thank you very much for your nice comments.

Reviewer 2 Report

Overall, a clear and concise systematic review of a very important topic. It is sad that we still know so little.  

11.  In the abstract and throughout the article, the vocabulary ”bipolar relapse” is used. Consider using only “relapse” or “affective episode relapse”.

P2. Page 1. Mixed episode. This nomenclature is no longer used. According to the DSM-5, all types of episodes can carry symptoms of the opposite pole, “mixed features”. In the DSM 5, “mixed episodes” was removed as a diagnostic entity and replaced with a “mixed specifier”.  Hence, it is not recommended to use “mixed episode” in new research.

33. Page 2. “Of all women diagnosed with postpartum depression, up to 54% were diagnosed with BD, but only 10% had a previous diagnosis [9]”. Vaguely cited, had to search the source of origin. Consider rewriting. “Of all women diagnosed with postpartum depression, 54% met the diagnostic criteria of BD, but only 10% of those had a previous diagnosis”. Further; “Therefore, it should be considered in those postpartum women with atypical depressive symptoms, with a family history of BD, or who do not respond to antidepressant treatment”. Consider rewriting. “Therefore, BD depression should be considered postpartum in women with atypical depressive symptoms…”

44.  Page 3. Method. Who read and approved excluded/included articles?

55. Page 7. Throughout. Either mood-stabilizers or mood stabilizers.

66. Page 7. Sociodemographic factors 3.5.1. ; a long paragraph in one sentence; difficult to follow and understand. Clinical factors 3.5.2; “having a longer duration of the same”. “The same”, what?

77. Page 8. “One study 215 points out that this risk is even higher after live birth compared to other reproductive events”. Difficult to understand. Consider rewriting. “….compared to miscarriage and induced abortion”.

88. Page 8. “TB diagnosis”à “BD diagnosis”?

99.  Discussion. Please add a more nuanced discussion regarding risks versus benefits of medication/relapse during pregnancy and the postpartum period.

Author Response

Overall, a clear and concise systematic review of a very important topic. It is sad that we still know so little.

Thank you very much for your nice comments.

  1. In the abstract and throughout the article, the vocabulary ”bipolar relapse” is used. Consider using only “relapse” or “affective episode relapse”.

 We have changed it in the whole manuscript.

 P2. Page 1. Mixed episode. This nomenclature is no longer used. According to the DSM-5, all types of episodes can carry symptoms of the opposite pole, “mixed features”. In the DSM 5, “mixed episodes” was removed as a diagnostic entity and replaced with a “mixed specifier”. Hence, it is not recommended to use “mixed episode” in new research.

 We have changed “mixed episodes” for “mixed features” in the whole manuscript.

 Page 2. “Of all women diagnosed with postpartum depression, up to 54% were diagnosed with BD, but only 10% had a previous diagnosis [9]”. Vaguely cited, had to search the source of origin. Consider rewriting. “Of all women diagnosed with postpartum depression, 54% met the diagnostic criteria of BD, but only 10% of those had a previous diagnosis”. Further; “Therefore, it should be considered in those postpartum women with atypical depressive symptoms, with a family history of BD, or who do not respond to antidepressant treatment”. Consider rewriting. “Therefore, BD depression should be considered postpartum in women with atypical depressive symptoms…”

We agree with the reviewer. We have changed these sentences.

  1. Page 3. Method. Who read and approved excluded/included articles?

 Thank you for your comment. We have included in the Study selection process:

The four phases were realized for two coauthors (JC and A S-G.)

  1. Page 7. Throughout. Either mood-stabilizers or mood stabilizers.

We have unified two mood-satbilizers in the whole manuscript.

  1. Page 7. Sociodemographic factors 3.5.1. ; a long paragraph in one sentence; difficult to follow and understand. Clinical factors 3.5.2; “having a longer duration of the same”. “The same”, what?

We have divided this paragraph in two sections.

We have explained it better:

Having a longer duration of the episode.

  1. Page 8. “One study 215 points out that this risk is even higher after live birth compared to other reproductive events”. Difficult to understand. Consider rewriting. “….compared to miscarriage and induced abortion”.

We agree with the reviewer. We have included it.

 Page 8. “TB diagnosis”à “BD diagnosis”?

We have changed it.

  1. Discussion. Please add a more nuanced discussion regarding risks versus benefits of medication/relapse during pregnancy and the postpartum period.

We have included a new paragraph in the Medication use section in the Discussion:

Analyzing the results of our review and of recent works that have studied the possible teratogenic effects of mood stabilizers, we believe that the most convenient thing to do is to continue using lithium treatment at the lowest possible dose (especially during the first trimester of pregnancy) [49] while valproic acid is completely contraindicated [50].

And two new references (number 49 and 50):

Fornaro, M., Maritan, E., Ferranti, R., Zaninotto, L., Miola, A., Anastasia, A., Murru, A., Solé, E., Stubbs, B., Carvalho, A.F., Serretti, A., Vieta, E., Fusar-Poli, P., McGuire, P., Young, A.H., Dazzan, P., Vigod, S.N., Correll, C.U., Solmi, M. Lithium Exposure During Pregnancy and the Postpartum Period: A Systematic Review and Meta-Analysis of Safety and Efficacy Outcomes. Am. J. Psychiatry. 2020; 177, 76-92. https://doi.org/10.1176/appi.ajp.2019.19030228

García-Portilla, M.P., Bobes, J. Preventive recommendations on the use of valproic acid in pregnant or gestational women to be very present. Rev. Psiquiatr. Salud Ment. 2017; 10, 129-133. https://doi.org/10.1016/j.rpsm.2017.06.001
